# Understanding Vaccine Hesitancy in Vietnamese Fish Farmers

**DOI:** 10.3390/antibiotics11070878

**Published:** 2022-06-30

**Authors:** Julie A. Chambers, Margaret Crumlish, David A. Comerford, Le-Hong Phuoc, Vo-Hong Phuong, Ronan E. O’Carroll

**Affiliations:** 1Department of Psychology, University of Stirling, Stirling FK9 4LA, UK; j.a.chambers@stir.ac.uk; 2Institute of Aquaculture, University of Stirling, Stirling FK9 4LA, UK; margaret.crumlish@stir.ac.uk; 3Department of Economics, University of Stirling, Stirling FK9 4LA, UK; david.comerford@stir.ac.uk; 4Research Institute for Aquaculture Number 2, 116 Nguyen Dinh Chieu Street, District 1, Ho Chi Minh City 700000, Vietnam; lehongphuoc@yahoo.com (L.-H.P.); vohongphuong@yahoo.com (V.-H.P.)

**Keywords:** antibiotic resistance, antimicrobial resistance, antibiotic stewardship, antimicrobial stewardship, animal, psychology, vaccination, aquaculture

## Abstract

(1) Background: Antibiotic (AB) usage in food animals is a significant contributor to antimicrobial resistance (AMR). Vaccination can reduce the over-use of AB treatment. Little is known about farmers’ attitudes and beliefs about AB and vaccine usage in developing countries, especially in aquaculture. (2) Methods: We used the necessity/concerns framework to guide our research, where vaccine hesitancy is viewed as a function of the perceived necessity versus the perceived concerns about treatment. We measured disease and treatment perceptions in 400 Vietnamese farmers of Pangasius catfish, specifically regarding (a) chemical treatment of water, (b) antibiotic usage, and (c) vaccination of fish. (3) Results: Although farmers’ concerns about AB usage outweighed necessity beliefs, 86.5% reported having used ABs on their farm. Knowledge and attitudes towards vaccination were positive, with views of its necessity outweighing concerns. However, if available, only 67.6% said they would definitely use vaccines in the future. Farmers were more likely to use vaccines if they reported having fewer problems with fish disease, felt that any concerns about vaccines were outweighed by their perceived benefits, had less mistrust of vaccination, and had fewer concerns about commercial profiteering. (4) Conclusion: Interventions that highlight concerns about continued antibiotic use, reduce concerns, and mistrust and increase the perceived necessity of vaccines combined with greater availability of vaccines may be the most effective way of overcoming vaccine hesitancy and increase appropriate use of vaccines by Vietnamese fish farmers.

## 1. Introduction

Antimicrobial resistance (AMR) is an escalating risk to global health in both animals and humans, with a recent review estimating that, in 2019, as many as 4.95 million deaths in people were associated with bacterial AMR worldwide, including 1.27 million deaths directly attributable to bacterial AMR [1]. Bacteria acquire antibiotic resistance through a diverse range of biological mechanisms, and may be acquired naturally over time, or driven through misuse or over-use of particular antibiotics. The ultimate outcome of any AMR development is treatment failure. AMs are classified as any treatment used to prevent and treat infections in humans, animals, and plants. The term AM includes antibiotics, disinfectants, antivirals, antifungals, and antiparasitics. Initiatives to tackle its threat to life by reducing the overuse and misuse of AMs are insufficient, especially in animal populations [2]. The latter has important implications for AMR, because the use of the same AMs or ABs in both human and food animal medicine means that AMR may spread from animals to humans [3]. Despite this, there is limited research on the behavioral influences of inappropriate use of AMs in food animal populations, particularly in the developing world [3]. Although the mechanisms which lead to AMR are biological, the motivations behind current levels and methods of antimicrobial (AM) usage are determined by a wide range of factors including individual, psychological, social, cultural, political, and economic forces.

There is a dearth of evidence on psychological determinants of behavior with regard to AM usage in all food animals [3,4]. Our recent review of determinants of AM usage in both humans and animals [5] found that awareness of AMR in farmers of both terrestrial and aquatic food animals was often poor, especially in developing countries [6,7]. Further, good knowledge of AMR did not necessarily translate into reduced AM usage [8]. Farmers often relied more on other farmers’ opinions, rather than advice from veterinarians, when deciding whether to use AMs in animals [9]. Economic factors and animal health were major determinants of farmers’ usage of AMs [8,10,11]. In addition, famers often had poor awareness of AMR and did not associate inappropriate usage of AMs in animals (such as to promote growth or to prevent rather than treat disease) with AMR in humans [12,13]. Most of the reviewed research was conducted with land-based animals in developed countries, including Europe and the US, e.g., [14,15,16,17], and there was little research in farmed fish or in developing countries [5]. This is despite the fact that large quantities of AMs are used in aquaculture in many developing countries, often without professional consultation or supervision [18].

In order to add to the knowledge base, the current research assesses attitudes towards treatment of disease amongst farmers of Pangasius catfish in Vietnam. This is an important area of research because aquaculture is the fastest growing food production sector worldwide [19], and Vietnam is one of the world’s largest producers of aquaculture products for human consumption, and the world leader in production of catfish in 2021, reaching 1.5 million tons [20]. The contribution of Vietnam to total world aquaculture production has increased from 3.24% in 2005 to 5.04% in 2018, and in 2020, the seafood export value of Vietnam reached 8.5 billion US dollars [21]. Fish health and welfare practices in the freshwater aquaculture sector vary between species produced, but generally most farmers rely on the use of antibiotics or other chemotherapeutants to prevent, treat, and manage disease prevalence in their production stocks. This results in an overuse and misuse of antibiotics.

A recent study conducted in Vietnam found that many farmers manage disease in their fish stock by unfavorable use of antibiotics, e.g., in the absence of clinical signs as a preventive measure [22]. Antibiotics are administered by fish farmers, usually mixed into food, and in Vietnam often without veterinary intervention [23]. Fish farmers in Vietnam also use non-medicated methods, e.g., herbal extracts and chemical treatment of water to help them manage disease [24,25]. An alternative to antibiotics—a vaccine—is available for two bacterial diseases causing disease in farmed Pangasius catfish, but little is known about how farmers regard this method of disease management. Understanding why Vietnamese catfish farmers may be reluctant to use vaccines is the primary aim of this study.

There is limited research regarding beliefs and attitudes towards AM usage in this population [5]. One study, which looked at farmers’ knowledge and opinions on antibiotic use in freshwater Vietnamese fish and shrimp farms (*n* = 94: 63 fish farms and 31 shrimp farms) who produce for the domestic market, concluded that farmers had poor knowledge of the purpose of using antibiotics on their farms [26]. Almost half (45%) of farmers interviewed did not believe antibiotics were effective in treating disease, but nonetheless most (72%) stated they used antibiotics on a regular basis either to treat or prevent disease. Those who did not use antibiotics reported their fish or shrimp were healthy so there was no need [26]. Farmers’ reasons for both using and not using antibiotics on their farms were more due to economic issues than any need to comply with regulations or due to any expressed concerns about AMR. A more recent study [27] examined knowledge and attitudes towards AMR in Vietnamese pig, poultry, and fish farmers (*n* = 392), and found the primary reported reason for AM usage was to treat infection. However, many farmers also gave misguided reasons for antibiotic use, including abnormal animal behavior, changing weather, or disease on adjacent farms. 

The objective of the current study is to examine the attitudes and beliefs of catfish farmers towards fish disease and its treatment, specifically (a) chemical treatment of water, (b) antibiotic usage, and (c) vaccination of fish, and also to assess farmers’ willingness to vaccinate fish as a means of disease prevention. We have used the necessity/concerns framework [28] to guide our research, where treatment hesitancy is viewed as a function of the perceived necessity of treatment versus perceived concerns about its use. In order to measure farmers’ attitudes and beliefs, we have adapted three robust measures of disease and treatment perceptions, which have shown to predict treatment adherence in humans [29]. 

## 2. Results

Cần Thơ province had a much higher percentage of growout only farms (89.1% versus 47.8% for all three provinces) and very few nursery-only farms (5.4% versus 49.0%); correspondingly, An Giang (60.7%) and Đồng Tháp (63.4%) provinces had more nursery only and hence fewer growout only farms. A chi-squared test of type of farm by region was highly statistically significant (χ^2^ (4) = 91.9, <0.001). Cần Thơ province also had fewer independent (78.3% versus 86.3%) and more contract (19.6% versus 9.8%) establishments overall, which was also statistically significant (χ^2^ (4) = 20.6, <0.001).

Participants appeared to have quite good knowledge of vaccine usage (Table 1), with over 90% reporting that they understood their usage in people (99.8%) and fish (92.5%), and 88% correctly identifying that vaccines worked via prevention. However, only 67.5% said they would use vaccines on their farm if they were available, with 25.8% being unsure and 6.8% saying they would not.

Beliefs about fish disease (BIPQ) are shown in Table 2. On average, participants reported some worries and problems with fish disease, but also believed that they had a degree of control over fish disease, that treatment helped, and that any disease would not last long. 

Table 3 shows the mean scores of the necessity and concerns subscales of the adapted BMQ measures regarding chemical treatment of water, using antibiotics in fish and vaccination of fish on their farms. This is the first time the BMQ has been adapted for attitudes towards disease in fish, and therefore confirmatory factor analysis was carried out on all 3 scales, as reported in Appendix A. The factor analysis supports a two-factor structure for each application of the BMQ measures, explaining between 43.4% and 45.9% of the variance in each case. Cronbach’s alpha values are shown in Table 3 and show low to moderate reliability. Removal of the item ‘(treatment)’ is a mystery to me resulted in improved reliability for the concerns subscale to CT, 0.40, AB, 0.55 and VAC, 0.56)—see Appendix A. A summary necessity minus concerns score was calculated for each treatment, where a positive score indicates that necessity beliefs outweigh concerns (Figure 1).

Paired sample t-tests show that the 0.04 difference between BMQ necessity and concerns for chemical treatment of water was not statistically significant. For attitudes towards antibiotic usage, concerns significantly outweighed necessity (mean difference −0.43, 95% CI (−0.51, −0.35)). In contrast, for attitudes towards vaccine usage, necessity significantly outweighed concerns (mean difference 0.37, 95% CI (0.30, 0.43)).

There were also differences between treatments with views of necessity minus concerns being higher for vaccines versus chemical treatment (mean difference 0.33, 95% CI (0.24, 0.41)) and antibiotics (mean difference 0.80, 95% CI (0.70, 0.90)). Views of necessity versus concerns for chemical treatment were also higher than those for antibiotics (mean difference 0.47, 95% CI 0.38, 0.56)). 

Table 4 shows the mean scores of the subscales of the VAX questionnaire. This is the first time the VAX has been adapted for attitudes towards disease in fish and so confirmatory factor analysis was carried out on the VAX items, results are reported in Appendix A. The factor analysis supports a four-factor structure, explaining between 43.4% and 45.9% of the variance in each case. Cronbach’s alpha values are shown in Table 4, and show moderately good reliability, with the exception of ‘concerns about commercial profiteering’. As mentioned in the Methods section, some items proved difficult to translate reliably into Vietnamese, which seemed to result in some misunderstandings of concepts. This particularly applied to the ‘concerns about profiteering’ items which may be reflected in this lower reliability. 

Figure 2 shows a comparison of scores across the VAX measure. It can be seen from the VAX subscales in Figure 2 that most concerns about vaccine usage were related to worry about unforeseen future effects, with the least concerns about vaccine mistrust.

### 2.1. Differences by Use of Treatments

All participants reported having used chemical treatment and none reported having used vaccination, so differences were examined (using one-way ANOVAs) by previous antibiotic use and by whether they would use vaccines in the future. There was no relationship between previous antibiotic use and whether they would use vaccines in the future (χ^2^ (2) = 2.2, *p* = 0.330).

Previous antibiotic use was related to farm type, with 94.4% (*n* = 185) of nursery only farms having used antibiotics, compared to 78.5% (*n* = 150) of growout only and 84.6% (*n* = 11) of combined farms. This difference was significant (χ^2^ (2) = 20.9, *p* < 0.001).

Table 5 shows significant differences on the attitude measures by previous use of ABS. A graph of these findings is shown in Figure 3. For those who had used antibiotics previously, significant effects were found for two BIPQ subscales. For ‘How much does fish disease on your farm affect your life’, those using antibiotics had significantly higher scores (i.e., reported a greater effect of fish disease) than those not using antibiotics (mean difference = 0.77). For ‘How long do you think fish disease will last on your farm’, those using antibiotics had significantly higher scores (i.e., believed fish disease would last longer) than those not using antibiotics (mean difference = 0.32).

For those who had used antibiotics previously (Table 5), significant effects were found for BMQ necessity of chemical treatment (mean difference = 0.17), with those using antibiotics seeing a greater necessity for chemical treatment. There was also a highly significant effect for necessity of antibiotics (mean difference = 0.42), with those who had previously used ABs reporting greater necessity. There was also a significant effect for necessity minus concerns re: antibiotics (mean difference = −0.35), with both groups reporting that concerns about AB use outweighed necessity, but this was greater in those who reported not having used ABs previously. These findings support the use of the BMQ Antibiotics scale as a reliable measure of attitudes towards AB use. There were no other significant effects on any of the attitudes measures for previous antibiotic use.

Table 6 shows significant differences on the attitudes measures by future use of vaccination. A graph of these findings is shown in Figure 4. For whether participants would use vaccination in the future, there was a significant effect for the BIPQ question ‘How much do you experience problems with fish disease’, with those who would use vaccines experiencing significantly *fewer* problems than those saying No (and neither significantly different from Not Sure). There were no significant differences by future vaccination use on any of the remaining BIPQ scales.

For whether participants would use vaccination in the future there were significant effects on all of the vaccine attitude subscales (Table 6 and Figure 4), with higher scores on necessity versus concerns, and more favorable attitudes towards vaccines in general in those who said Yes or Not Sure versus No. There were no significant effects on any of the subscales regarding attitudes towards chemical treatment or antibiotic use. 

The results in Table 6 strongly confirm the robustness of the BMQ VAC and the VAX scales as measuring attitudes towards vaccine use in fish in this population.

### 2.2. Relationships between Attitude Measures

Appendix A shows correlations between the measures of attitudes, the findings for significant correlations (chosen as *p* < 0.01, due to the number of tests) are summarized below.

#### 2.2.1. Illness Beliefs (BIPQ)

Farmers who reported they were more affected by fish disease also reported that they had less control (r = 0.16, *p* = 0.001), more problems (r = 0.57, *p* < 0.001), more worries (r = 0.60, *p* < 0.001), and more emotional worries (r = 0.58, *p* < 0.001) about fish disease. They also had a lower belief in the necessity of chemical treatment (r = −0.16, *p* = 0.001) and were less likely to believe that necessity outweighed concerns for chemical treatment (r = −0.15, *p* = 0.003). 

Farmers who reported they were more likely to be able to control fish disease reported higher beliefs in the helpfulness of treatment (r = 0.47, *p* < 0.001) and a better understanding of fish disease (r = 0.32, *p* < 0.001). 

Farmers reporting higher beliefs in the helpfulness of treatment also reported a better understanding of fish disease (r = 0.32, *p* < 0.001). 

Farmers reporting more problems from fish disease also reported more worries (r = 0.63, *p* < 0.001) and emotional effects (r = 0.49, *p* < 0.001) of fish disease. They also had lower views of the necessity of vaccination (r = −0.18, *p* < 0.001) and the necessity of chemical treatment (r = −0.23, *p* < 0.001) and were less likely to believe that necessity outweighed concerns for chemical treatment (r = −0.27, *p* < 0.001). They also had higher concerns about the use of both chemical treatment (r = 0.14, *p* = 0.005) and antibiotics (r = 0.16, *p* = 0.002). 

Farmers reporting more worries about fish disease also reported more emotional effects (r = 0.70, *p* < 0.001). Similar to problems, there were relationships for general worries about fish disease with views of less necessity for chemical treatment (r = −0.20, *p* < 0.001), and also their concerns about chemical treatment outweighed their views of its necessity (r = −0.18, *p* < 0.001). 

Farmers reporting a better understanding of fish disease had fewer concerns about vaccination (r = −0.14, *p* = 0.004). 

Farmers reporting more emotional effects also had lower beliefs in the necessity of chemical treatment (r = −0.27, *p* < 0.001) and vaccines (r = −0.16, *p* = 0.001) and a stronger belief that concerns about chemical treatment outweighed necessity (r = −0.22, *p* < 0.001). 

None of the BIPQ items were related to any of the VAX subscales at *p* < 0.01.

#### 2.2.2. Beliefs about Medicines (BMQ)

Beliefs about the necessity of chemical treatment were related to believing in the necessity of both antibiotics (r = 0.32, *p* < 0.001) and vaccines (r = 0.18, *p* < 0.001) and was also related to believing that the necessity of using both chemical treatment (r = 0.77, *p* < 0.001) and antibiotics (r = 0.17, *p* < 0.001) outweighed any concerns. Similarly, concerns about chemical treatment were strongly related to concerns about both antibiotics (r = 0.46, *p* < 0.001) and vaccines (r = 0.23, *p* < 0.001), and also to the belief that concerns outweighed necessity for all three treatments (AB: (r = −0.26, *p* < 0.001), chemical treatment: (r = −0.65, *p* < 0.001), and vaccination: (r = −0.21, *p* < 0.001)). 

Higher views of the necessity of antibiotics were negatively related to concerns about their use (r = −0.18, *p* = 0.006) and to their necessity outweighing concerns (r = 0.74, *p* < 0.001). Higher views of the necessity of vaccines were related to lower concerns about their use (r = −0.37, *p* < 0.001), and higher necessity minus concerns (r = 0.83, *p* < 0.001). 

#### 2.2.3. Attitudes towards Vaccination (VAX)

All VAX subscales were highly correlated with each other in the expected direction and also with the BMQ VAC Necessity and Concerns scales (all *p* < 0.001, except VAX preference for natural immunity and BMQ vaccine necessity (r = −0.13, *p* = 0.007) and VAX mistrust (r = 0.14, *p* = 0.006)), reinforcing the validity of the measure in this population.

### 2.3. Attitudes by Types of Establishment and Region

There were no differences by farm role (owner, worker, manager) on any of the attitude measures. The results of one-way ANOVAs by type of farm (nursery/growout), type of farm ownership (company/contract/independent), and region are shown in Appendix A. 

### 2.4. Intended Vaccination Use by Farm Type Farm Ownership and Province

Chi-squared tests were used to examine the relationship between future vaccine usage (No, Not sure, Yes) and farm type, farm ownership, and province. There were no effects for farm role (χ^2^ (4) =1.2, *p* = 0.884), farm type (χ^2^ (4) = 4.8, *p* = 0.311), or ownership (χ^2^ (4) = 4.4, *p* = 0.361). There was a significant difference for province (χ^2^ (4) = 10.5, *p* = 0.032), with those from An Giang being more likely to say No (10.4% versus 6.8% overall) and less likely to say Yes (60.1% versus 67.5%), and those from Đồng Tháp being more likely to say Yes (74.5% versus 67.5%). This was despite those from Đồng Tháp reporting less favorable views of vaccination (Appendix A).

### 2.5. Predictors of Future Vaccine Usage

An ordinal regression was conducted on future vaccine usage (No/Not Sure/Yes) with the eight disease beliefs items (BIPQ), the three treatment beliefs (BMQ necessity minus concern items for chemical treatment, antibiotic use, and vaccination), the four vaccination attitudes (VAX) subscales and province as predictor variables (the latter as a categorical variable (Farm role, type, and ownership were not included as these were not related to future vaccine usage)). The necessity minus concerns items were chosen in preference to individual necessity and concerns subscales as predictor variables are advised not to be highly correlated. Results for significant predictors (*p* < 0.05) are shown in Table 7. 

There were highly significant results on three of the vaccination measures (all *p* < 0.001): farmers were more likely to use vaccines in the future if they had higher scores on necessity minus concerns for vaccines (i.e., necessity outweighed concerns: approximately 2.21 times more likely for each 1 point increase on this measure), had less mistrust of vaccines (approximately 6.89 times more likely for each 1 point decrease on this measure), and had fewer concerns about commercial profiteering re: vaccines (approximately 2.46 times). There were also some lesser effects: farmers were likely to be more positive about vaccine use if they experienced fewer problems with fish disease (approximately 1.25 times more likely for each 1 point decrease on this measure), had lower necessity minus concerns scores for antibiotics (i.e., approximately 1.46 times more likely to say they would use vaccines for each 1 point decrease on necessity minus concerns for ABs), and were from Đồng Tháp versus An Giang province (approximately 3.28 times) or Đồng Tháp versus from Cần Thơ province (approximately 3.01 times).

## 3. Discussion

The current study adapted three robust measures of illness and treatment beliefs in humans in order to examine the beliefs and attitudes of Vietnamese farmers of Pangasius catfish towards fish disease and its treatment by chemicals, ABs, and vaccination. The revised measures of illness beliefs (BIPQ), medication beliefs (BMQ), and attitudes towards vaccination (VAX) scales showed good validity and moderate reliability in this group. Reliability was improved by the removal of items whose concept had proved difficult to translate from English into Vietnamese, likely due to differences in cultures. 

On the measure of illness/disease beliefs (BIPQ), farmers who reported experiencing more problems with fish disease also reported more worries and emotional impacts on their lives. Those who reported being more affected by fish disease also had a lower belief in the necessity of chemical treatment. Farmers reporting more problems from fish disease had higher concerns about the use of antibiotics, as well as a lower belief in the necessity of vaccines and chemical treatment. In contrast, farmers reporting a better understanding of fish disease also had higher beliefs in the helpfulness of treatment, reported fewer problems and worries regarding fish disease, and had fewer concerns about vaccination. 

Overall, farmers’ concerns about AB usage significantly outweighed their views of their necessity. In addition, farmers reporting more problems from fish disease also had higher AB concerns, suggesting a good awareness of issues surround AB usage. Nonetheless, ABs had been used by the vast majority of farmers (88%). Thus, it would seem that concerns about using ABs to treat fish did not result in reduced reliance on ABs in these farmers. Other research in Vietnamese food farmers also found that farmers view ABs as the first line of defense (including using ABs to prevent disease), despite good knowledge and awareness of the risks of AB usage, and often believing they are ineffective [22,26,27]. Reasons for this may include perceived lack of alternative treatments (including lack of awareness of vaccination), lack of a therapeutic approach (including the absence of rapid, cost-effective diagnostics and the switching to another AB if the first one fails [22,26]), lack of infrastructure to control access/availability of antibiotics and alternative treatments, and the view that as AB usage is universal then there must be good reason for its usage. 

In addition, farmers are likely to view the current vaccine as inconvenient, as it is often delivered by injection, and they may not yet be aware of the benefits of vaccination in terms of disease prevention. We are planning a follow-up qualitative study to explore these hypotheses in greater detail. Those who had used ABs also reported experiencing more effects of fish disease, expecting fish disease to last longer, and had a greater view of the necessity versus concerns of using both ABs and chemical treatment than farmers who had not used ABs. This suggests that farmers who used ABs viewed both ABS and chemical treatment of water, rather than vaccination, as essential in treating fish disease.

No published research examining attitudes of fish farmers towards vaccinating fish was found. In the current study, knowledge of and attitudes towards vaccination were quite positive, with 88% correctly identifying that vaccines worked by preventing disease and overall views of the necessity of vaccines significantly outweighing any concerns. Nonetheless only 67.6% said they would definitely use vaccines in the future if they were available, with 7% saying they would not and the remainder being unsure. A recent study with cattle farmers in India [30] also found positive attitudes towards vaccination and good understanding of its purpose, and that practical issues (such as difficulty of accessing vaccination programs) were more strongly related to lack of uptake than fear of adverse reactions. They also found that knowledge gleaned from professionals such as veterinarians increased uptake and suggested that farmer education was required before immunization programs were launched. In the current study, farmers in Đồng Tháp gave more favorable responses towards vaccination than the other two provinces. Possible explanations for this are that farmers in Đồng Tháp have received more training on fish disease and its treatment from the sub-department of fisheries, and that we have conducted prior research in this province.

Farmers were less likely to say they would use vaccines if they had more problems with fish disease, more negative general views of vaccination, more concerns about using vaccines on their own farm, and a stronger view that necessity outweighed concerns for AB usage. Lindahl et al. [31] found that poultry farmers in Kenya and Tanzania were more likely to take up vaccine usage if they had knowledge of what a vaccine does and when they lived in villages which were involved in a support program for vaccine usage. Another study in Tanzania [32] found that knowing other poultry farmers who had used vaccines increased the likelihood of vaccine usage. Thus, support from both professionals and peer groups may influence willingness to use vaccines in preference to ABs.

Although farmers generally expressed concerns about AB usage and had positive views of vaccination, overall this did not translate into being certain they would use vaccines in the future if they were available. As part of our research, a choice experiment was also conducted with the same farmers [33]. This indicated that, with favorable conditions, including high effectiveness of vaccine, realistic cost, and appropriate method of administration (80% effective, treats both red and white spot, delivery by bath, price 500,000 dong per 1000 fish), as many as 83.5% of the same farmers would use a vaccine. In a recent review of the evidence regarding vaccine hesitancy in humans, Brewer et al. [34] concluded that changing attitudes can lead to improving vaccination uptake but that facilitating vaccination directly, including removing barriers and easing access to the vaccination process may be more effective at changing vaccination behavior. Our findings suggest that interventions that highlight concerns about continued antibiotic use, reduce concerns and mistrust of vaccines (particularly concerns about commercial profiteering), and increase the perceived necessity of vaccines, combined with greater familiarity and availability of vaccines, may be the most effective way to overcome vaccine hesitancy and increase appropriate use of vaccines by Vietnamese fish farmers.

## 4. Materials and Methods

### 4.1. Participants

Interviews were conducted with 400 participants from catfish farms in three Vietnamese provinces: An Giang, Cần Thơ, and Đồng Tháp. The data were collected by trained researchers in face-to-face interviews between July and December 2020. Ethical approval was given by the University of Stirling General University Ethics Panel on 24 February 2020 (no. 821). Farmers were recruited to a study to help us understand factors that influence willingness to use vaccines in fish. 

In total, the An Giang province has 200 growout and 1100 nursery farms, Đồng Tháp has 160 growout and 1050 nursery farms, and Cần Thơ has 279 growout and 61 nursery farms.

The majority of interviewees were farm owners (86.3%, *n* = 345), 13.3% (*n* = 53) were farm managers, and 0.5% (*n* = 2) were farm workers. Almost half (49.0%, *n* = 191) of farms were nursery only, 47.8% (*n* = 191) were growout only, and 3.3% (*n* = 13) were both nursery and growout. Nursery farms hold fish from hatching to a weight of around 30 g per fish. Growout farms hold fish from around 30g to harvesting at around 1000 g per fish. Most (86.3%, *n* = 345) were independent concerns (i.e., owned and operated by independent farmers), 9.8% (*n* = 39) were contract (i.e., had a contract to run the farm on behalf of another owner), and 4% (*n* = 16) were company owned. 

Two-fifths (40.8%, *n* = 163) of farms were in An Giang province, 36.3% (*n* = 145) were in Đồng Tháp, and 23% (*n* = 92) were in Cần Thơ. An Giang is a province in southern Vietnam, bordering Cambodia to the north, and lying in the Mekong Delta. Đồng Tháp is also a province in the Mekong Delta and lies immediately east of An Giang, and also borders Cambodia to the north. Cần Thơ is a city and province-level municipality in southern Vietnam, situated on the left bank of the Hau Giang River. It lies to the south of both An Giang and Đồng Tháp.

All 400 interviewees had previously used chemical treatment of water on their farms; 86.5% (*n* = 346) had used antibiotics at some point (either in the past or currently), and none had previously used vaccines. 

### 4.2. Outcome Measures

The *Brief Illness Perceptions Inventory* (BIPQ) [35] was adapted to assess farmers’ perceptions and attitudes towards fish disease on their farm. The BIPQ is a psychometrically robust measure which has been used extensively to assess illness perceptions in humans. Questions were adapted to assess farmer’s attitudes to disease in their fish. Eight items (each measured on a scale of 0–10) were used, e.g., “Do you think you are able to control fish disease on your farm?” scored from 0 = “absolutely no control” to 10 = “definitely can control”. 

The *Beliefs about Medications Questionnaire* (BMQ) [36] was adapted to assess farmers’ beliefs about treatment of fish in their farm. The BMQ is a psychometrically robust instrument that reliably measures participants’ beliefs about the necessity and concerns of specific medicines in humans [28]. The same eight items were adapted to measure farmers’ attitudes towards different treatments in fish on their farm, i.e., (a) the use of chemical treatment; (b) the use of antibiotics; and (c) the use of vaccination (24 items in total). Example items include: “Chemical treatment of the water/antibiotics/vaccination protect(s) the fish on my farm from becoming diseased” (necessity) and “I sometimes worry about the long-term effects of using, chemical treatments of the water/antibiotics in the fish/vaccines in the fish, on my farm” (concerns). Items were scored on a 5-point Likert-type scale from “Strongly disagree” to “Strongly agree”. Necessity and concerns scores were calculated as means of the 4 individual items for each scale (necessity: items 1, 3, 6, and 7; concerns items: 2, 4, 5, and 8).

The *VAX* [37] scale was used to measure farmers’ beliefs associated with reluctance to use vaccines in general. It consists of 4 subscales: (1) Trust of vaccines; (2) Worries about unforeseen future events; (3) Concerns about commercial profiteering; and (4) Preference for natural immunity. The psychometric robustness of the scale has been recently confirmed [38]. VAX scores are correlated with medical mistrust, beliefs about medicines, and successfully differentiate vaccine users from non-vaccine users in people [35]. The final scale consisted of eight items which assessed farmers’ attitude towards vaccination in general.

All items on the above questionnaires were written in English, translated into Vietnamese, and backtranslated into English in an iterative process until correct interpretation of each item on each scale was assured. The questionnaire (see Appendix A) was pilot tested with ten participants for understanding, and resulting changes were made as required. This led to the removal of 4 items from the VAX, as cultural differences meant that certain items could not be translated in a reliable manner and/or participants were reluctant to answer the question (e.g., “Authorities promote vaccination for financial gain, not for the health of farmed fish”). The remaining 8 items represented 2 items from each subscale. Questionnaires were administered via face-to-face interview to all participants. 

### 4.3. Data Entry and Analysis

Questionnaire data was entered onto a Microsoft Excel 2010 spreadsheet which made use of data restrictions and drop-down menus to aid integrity of data entry. The resulting file was uploaded into SPSS v.28 for data analysis.

## Figures and Tables

**Figure 1 antibiotics-11-00878-f001:**
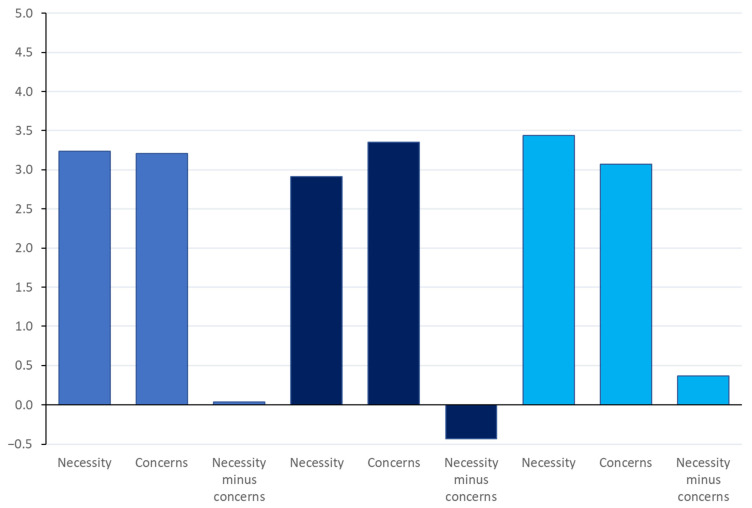
Mean scores on the Beliefs about Medication questionnaire.

**Figure 2 antibiotics-11-00878-f002:**
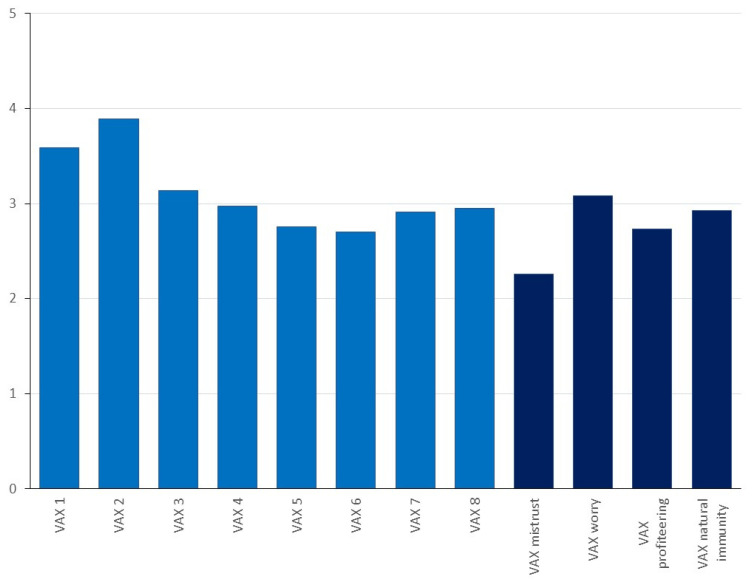
Mean scores on the VAX items (lighter blue) and VAX subscales (darker blue). Note: All items are scored on a range of 0 to 5; for the subscale calculation only, items 1–2 have been reverse-coded such that higher scores on all subscales indicate less favorable views of vaccination. VAX 1 = I feel that farmed fish are safe after being vaccinated; VAX 2 = I feel that farmed fish are protected after getting vaccinated; VAX 3 = Although most vaccines in farmed fish appear to be safe, there may be problems that we do not know about at present; VAX 4 = Vaccines can cause unforeseen problems in farmed fish; VAX 5 = Vaccines make a lot of money for pharmaceutical companies, but do not do much for regular fish farmers; VAX 6 = I do not think vaccination programs are honest or trustworthy; VAX 7 = Vaccines only work for a short time, while natural resistance to disease in farmed fish lasts longer; VAX 8 = Building up natural resistance to disease in farmed fish is safer than vaccination.

**Figure 3 antibiotics-11-00878-f003:**
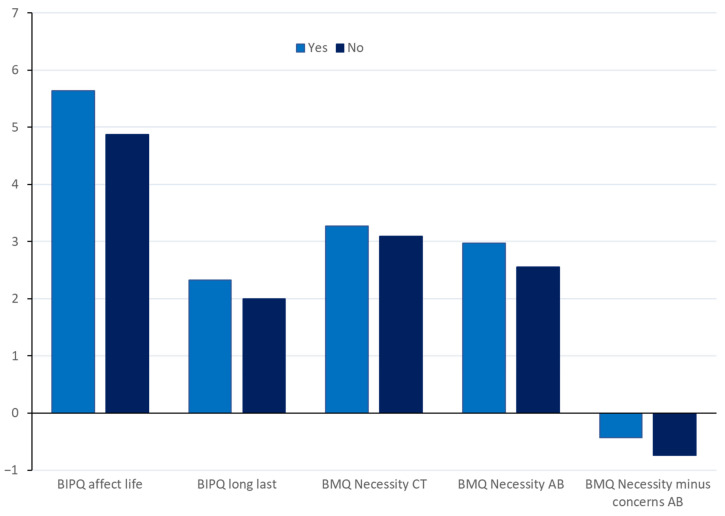
Significant results on attitudes measures by previous AB usage.

**Figure 4 antibiotics-11-00878-f004:**
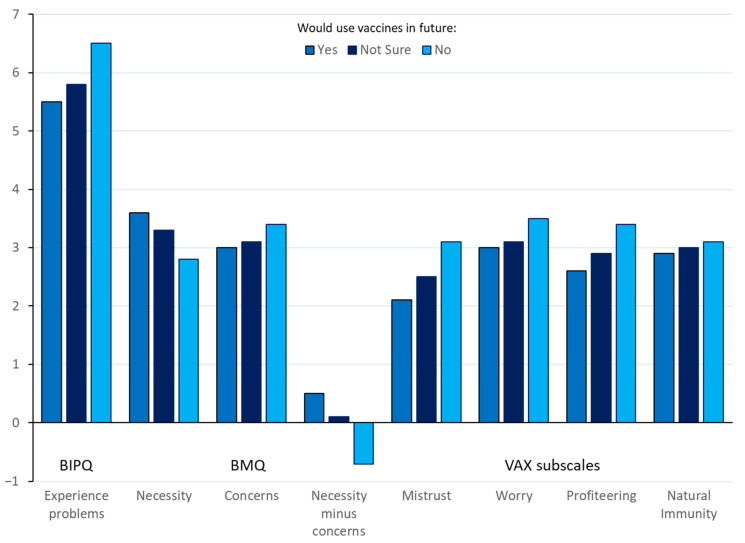
Significant differences on attitudes measures by future vaccine usage.

**Table 1 antibiotics-11-00878-t001:** Views of vaccine usage (*n* = 400).

Question	Yes	No	Not Sure	
Understand vaccine use in people	399 (99.75%)	1 (0.25%)	0 (0%)	
Know vaccines used in fish	370 (92.5%)	20 (5.0%)	10 (2.5%)	
Would use vaccines on your farm if they were available	270 (67.5%)	27 (6.8%)	103 (25.8%)	
	**Treat**	**Prevent**	**Both**	**Not sure**
How do vaccines work?	1 (0.25%)	352 (88.0%)	22 (5.5%)	25 (6.25%)

**Table 2 antibiotics-11-00878-t002:** Attitudes towards disease on their fish farm (*n* = 400).

Brief Illness Perceptions Questionnaire Item (Illness Belief)	Mean (S.D.)	Median (Range)
How much does fish disease on your farm affect your life? (*Consequences*)	5.53 (1.96)	6 (1–10)
How long do you think fish disease on your farm will last in the first production cycle (in weeks)? (*Timeline*)	2.28 (0.74)	2 (1–5)
Do you think you are able to control fish disease on your farm? (*Personal control*)	6.25 (1.61)	7 (1–10)
How much do you think treatment can help fish disease on your farm? (*Treatment control*)	6.78 (1.43)	7 (2–10)
How much do you experience problems from fish disease on your farm? (*Identity*)	5.62 (1.77)	6 (1–10)
How worried are you about fish disease on your farm? (*Disease concern*)	5.76 (2.06)	6 (1–10)
Do you think you have a good understanding of fish disease on your farm? (*Disease coherence*)	7.11 (1.42)	7 (2–10)
How much does fish disease on your farm affect you emotionally? (*Emotional impact*)	5.26 (2.19)	5 (0–10)

Note: All items scored on a range of 0 to 10, mid-point = 5.

**Table 3 antibiotics-11-00878-t003:** Necessity and concerns of different methods of treating disease on their fish farm (*n* = 400).

Beliefs about Medication (Treatment) Questionnaire	Mean (S.D.)	Reliability (Cronbach’s Alpha)
Necessity of chemical treatment	3.24 (0.53)	0.66
Concerns about chemical treatment	3.21 (0.44)	0.34
Necessity minus concerns about chemical treatment	0.04 (0.69)	
Necessity of use of antibiotics	2.91 (0.52)	0.63
Concerns about use of antibiotics	3.35 (0.53)	0.52
Necessity minus concerns about use of antibiotics	−0.43 (0.79)	
Necessity of vaccination of fish	3.44 (0.41)	0.51
Concerns about vaccination of fish	3.07 (0.41)	0.54
Necessity minus concerns about vaccination of fish	0.37 (0.68)	

Note: All items scored on a range of 0 to 5.

**Table 4 antibiotics-11-00878-t004:** Attitudes towards vaccine usage in fish (*n* = 400).

VAX Questionnaire (Subscale)	Mean (S.D.)	Reliability (Cronbach’s Alpha)
I feel that farmed fish are safe after being vaccinated—(*Mistrust*).	3.59 (0.57)	
I feel that farmed fish are protected after getting vaccinated—(*Mistrust*).	3.89 (0.67)	
Although most vaccines in farmed fish appear to be safe, there may be problems that we do not know about at present—(*Worry*).	3.14 (0.66)	
Vaccines can cause unforeseen problems in farmed fish—(*Worry*).	2.97 (0.66)	
Vaccines make a lot of money for pharmaceutical companies, but don’t do much for regular fish farmers—(*Profiteering*).	2.76 (0.73)	
I do not think vaccination programs are honest or trustworthy—(*Profiteering*).	2.70 (0.69)	
Vaccines only work for a short time, while natural resistance to disease in farmed fish lasts longer—(*Natural immunity*).	2.91 (0.65)	
Building up natural resistance to disease in farmed fish is safer than vaccination—(*Natural immunity*).	2.95 (0.63)	
**Subscales**		
Mistrust of vaccine benefit	2.26 (0.53)	0.63
Worry over unforeseen future effects	3.06 (0.60)	0.77
Concerns about commercial profiteering	2.73 (0.55)	0.34
Preference for natural immunity	2.93 (0.54)	0.62

Note: All items scored on a range of 0 to 5; for the subscale calculation only, items 1–2 have been reverse-coded such that higher scores on all subscales indicate less favorable views of vaccination.

**Table 5 antibiotics-11-00878-t005:** Significant differences on BIPQ and BMQ measures by previous use of ABs (*n* = 400).

Subscale	F (1, 398), *p* Value	Yes (Y)	No (N)	95% CI for Mean Diff
BIPQ How much does fish disease affect your life	7.2, *p* = 0.008	5.64 (1.94)	4.87 (2.03)	(0.21, 1.33)
BIPQ How long do you think fish disease will last	8.8, *p* = 0.003	2.32 (0.72)	2.00 (0.82)	(0.11, 0.53)
BMQ Necessity of chemical treatment	5.1, *p* = 0.024	3.27 (0.52)	3.09 (0.53)	(0.02, 0.32)
BMQ Necessity of ABs	33.5, *p* < 0.001	2.97 (0.51)	2.55 (0.42)	(0.28, 0.56)
BMQ Necessity minus concerns for ABs	9.4, *p* = 0.002	−0.43 (0.79)	−0.74 (0.77)	(0.13, 0.58)

**Table 6 antibiotics-11-00878-t006:** Significant differences on attitudes scales by future use of vaccine (*n* = 400).

Subscale	F (2397), *p* Value	Yes (Y)	Not Sure (NS)	No (N)	Scheffe Post Hoc Diffs
BIPQ: How much do you experience problems with fish disease	5.1, *p* = 0.007	5.5 (1.77)	5.8 (1.81)	6.5 (1.31)	Y < N
BMQ Necessity of vaccines	79.6, *p* < 0.001	3.6 (0.34)	3.3 (0.38)	2.8 (0.30)	Y > NS > N
BMQ Concerns over vaccines	14.9, *p* < 0.001	3.0 (0.40)	3.1 (0.41)	3.4 (0.24)	Y, NS < N
BMQ Necessity minus concerns	60.5, *p* < 0.001	0.5 (0.58)	0.1 (0.67)	−0.7 (0.46)	Y > NS > N
VAX mistrust of vaccines	81.2, *p* < 0.001	2.1 (0.42)	2.5 (0.54)	3.1 (0.25)	Y < NS < N
VAX worries over future use	7.8, *p* < 0.001	3.0 (0.60)	3.1 (0.56)	3.5 (0.54)	Y, NS < N
VAX concerns over profiteering	40.8, *p* < 0.001	2.6 (0.47)	2.9 (0.59)	3.4 (0.45)	Y < NS < N
VAX preference for natural immunity	5.2, *p* = 0.006	2.9 (0.58)	3.0 (0.42)	3.1 (0.49)	Y < N

Note: Higher scores on the VAX subscales indicate less favorable views of vaccination.

**Table 7 antibiotics-11-00878-t007:** Significant predictors (*p* < 0.05) of future vaccine usage in ordinal regression analysis (*n* = 400).

	No versus Not Sure versus Yes				
			95% CI for Exp B		
Measure	B	Exp B	Lower	Upper	*p*
BIPQ 5 ^a^	−0.22 (0.10)	0.80	0.65	0.98	0.030
AB Nec-Conc ^a^	−0.38 (0.17)	0.68	0.49	0.96	0.028
VAC Nec-Conc	0.79 (0.27)	2.21	1.30	3.77	<0.001
VAX mistrust ^a^	−1.91 (0.33)	0.15	0.08	0.28	<0.001
VAX profit ^a^	−0.90 (0.28)	0.41	0.23	0.70	<0.001
Province D vs. A ^a^	−1.19 (0.34)	0.31	0.16	0.60	<0.001
Province D vs. C ^a^	−1.11 (0.44)	0.33	0.14	0.78	0.012
Nagelkerke r^2^	0.462				
χ^2^ (17)	183.6, *p* < 0.001				

Note: BIPQ 5 = How much do you experience problems from fish disease on your farm? AB Nec-Conc = Necessity minus concerns re: antibiotics; VAC Nec-Conc = Necessity minus concerns re: vaccines; VAX mean = mean of all 8 VAX items, VAX mistrust = the mistrust of vaccines subscale and VAX profit = the worries about commercial profiteering regarding vaccines subscale, higher scores represent less favorable views of vaccines on all VAX measures; Province D vs. A = Đồng Tháp versus An Giang; Province D vs. C = Đồng Tháp versus Cần Thơ. ^a^ As B is negative, the CI suggests that respondents were 1/Exp(B) times more likely to move up a category (from No to Not Sure or Not sure to Yes) for each one-point decrease on the relevant measure.

## Data Availability

The data presented in this study are openly available in OSF Storage at https://doi.org/10.17605/osf.io/bxz8d (accessed on 24 May 2022).

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
