# Peer review of "Understanding Vaccine Hesitancy in Vietnamese Fish Farmers"

_antibiotics, 2022, doi:10.3390/antibiotics11070878_

Round 1

Reviewer 1 Report

There is no special designation. I think it would be an important survey results analysis, but I feel it lacks a bit of scientific impact.

Author Response

There is no special designation. I think it would be an important survey results analysis, but I feel it lacks a bit of scientific impact.

Response: We trust the changes made in response to reviewers 2 and 3 address this comment.

Reviewer 2 Report

see gramatical mistakes

Author Response

English language and style are fine/minor spell check required - see gramatical mistakes

Response: We have now completed a grammatical check and made all necessary changes. Additional references have now been included – see response to Reviewer 3 below.

Reviewer 3 Report

Introduction.

Please note that the term antimicrobials also includes disinfectants. Is that the intension of the authors? Please clarify.

Line 76. …. Use of using antibiotics. Please correct.

Last paragraph. Please define clearly the objectives of the study.

Procedures

How many catfish farmers are in Vietnam?

I suggest to omit the data from 2 farm workers.

Please include the questionnaire used in a supplementary table.

Results

Tables 3 and 4. Please add graphs of these results (additionally to the table) to make understanding easier for future readers.

2.1. and 2.2. The text is difficult to follow. Please present detailed results in tables and briefly summarize in the text. Moreover, please use graphs.

Discussion

This is short and really shallow. The authors did not go into adequate depth into comprehending their own results. The discussion must be significantly extended and more references must be added.

Overall. The manuscript can be considered again for possible publication after implementation of the corrections indicated above.

Author Response

Introduction.

Please note that the term antimicrobials also includes disinfectants. Is that the intension of the authors? Please clarify.

Response:  Much of use of the term antimicrobial(s) is our report is in reference to published research, and many of these papers have not defined their usage of the term. The World Health Organisation defines AMs as medicines used to prevent and treat infections in humans, animals and plants including antibiotics, antivirals, antifungals and antiparasitics and we have included this explanation in a footnote on page 1.

Line 76. …. Use of using antibiotics. Please correct.

Response:  Corrected.

Last paragraph. Please define clearly the objectives of the study.

Response: we have changed the last paragraph of the Introduction so it now reads: “The objective of the current study is to examine the attitudes and beliefs of catfish farmers towards fish disease and its treatment, specifically (a) chemical treatment of water, (b) antibiotic usage and (c) vaccination of fish, and also to assess farmers’ willingness to vaccinate fish as a means of disease prevention. We have used the necessity/concerns framework [28] to guide our research, where treatment hesitancy is viewed as a function of the perceived necessity of treatment versus perceived concerns about its use. In order to measure farmers’ attitudes and beliefs, we have adapted three robust measures of disease and treatment perceptions, which have shown to predict treatment adherence in humans [29].”

Procedures

How many catfish farmers are in Vietnam?

Response: We have now added details of the catfish farms in the three areas under study in line 425 i.e., In total, the An Giang province has 200 growout and 1,100 nursery farms, Đồng Tháp has 160 growout and 1,050 nursery farms and Cần Thơ has 279 growout and 61 nursery farms.

I suggest to omit the data from 2 farm workers.

Response: As we do not examine data separately for the farm workers (compared to owners/managers), and omitting them does not affect the findings, we prefer to include these results for completeness.

Please include the questionnaire used in a supplementary table.

Response: The questionnaire is now included in supplementary materials, as Appendix A.

Results

Tables 3 and 4. Please add graphs of these results (additionally to the table) to make understanding easier for future readers.

Response: graphs are now added as Figures 1 and 2.

2.1. and 2.2. The text is difficult to follow. Please present detailed results in tables and briefly summarize in the text. Moreover, please use graphs.

Response: As there is a lot of data resulting from our study, we feel a table that attempts to describe all results in detail would overload readers and obscure those results that are most insightful. As such, we prefer to present only the most meaningful results in the text and use supplementary materials for tables which contain a lot of results which do not add greatly to the findings. However, we have included extra tables and graphs as requested by reviewer 3 as outlined below. We have also added further explanation to and/or simplified the text, where appropriate.

2.1: A table for the first part in 2.1 – differences by previous use of antibiotics - is added (Table 5). The second part – differences by future vaccination usage on the vaccine measures - already contains a table of the results (Table 6) which are then explained in the text. We have now added the significant finding on the BIPQ to this table. We have also included graphs for all significant findings in this section (Figures 3 and 4).

2.2: This section of correlations between attitudes measures is already included in a very large table – in supplementary materials Table S4. We feel that the corresponding scatterplots of these correlations, whilst being somewhat cumbersome, would not add greatly to the interpretation of the findings.

Discussion

This is short and really shallow. The authors did not go into adequate depth into comprehending their own results. The discussion must be significantly extended and more references must be added.

Response: We have now made extensive additions (including references) to the discussion which we hope addresses this reviewer’s concerns. In particular we have added more explanation to the paragraph between lines 351 and 364, regarding the fact that having higher concerns about using ABs does not necessarily translate into more appropriate usage of ABs. In the section between lines 374 and 401 we have considered attitudes to vaccination in the current study in reference to other published research.

Overall. The manuscript can be considered again for possible publication after implementation of the corrections indicated above.

Response: We trust that the changes made address the concerns of the reviewers.

Round 2

Reviewer 3 Report

Please note that the term antimicrobials also includes disinfectants. Is that the intension of the authors? Please clarify.

Response:  Much of use of the term antimicrobial(s) is our report is in reference to published research, and many of these papers have not defined their usage of the term. The World Health Organisation defines AMs as medicines used to prevent and treat infections in humans, animals and plants including antibiotics, antivirals, antifungals and antiparasitics and we have included this explanation in a footnote on page 1.

The response of the authors to the initial comment is vague and really neither offered any explanation, nor it improved the passage, which still remains badly written.
Please correct and please write in correct English language and please answer the question set initially.
After appropriate correction, the manuscript can be accepted.

Author Response

We note this reviewer’s concerns, and have now added, ‘disinfectants’ to the footnote on page 1. This now reads: “AMs are classified as any treatment used to prevent and treat infections in humans, animals and plants. The term AM includes antibiotics, disinfectants, antivirals, antifungals and antiparasitics.”